# Locus-specific histone deacetylation using a synthetic CRISPR-Cas9-based HDAC

Deborah Y. Kwon[1], Ying-Tao Zhao[1], Janine M. Lamonica[1] & Zhaolan Zhou[1]

Efforts to manipulate locus-specific histone acetylation to assess their causal role in gene expression and cellular and behavioural phenotypes have been impeded by a lack of experimental tools. The Cas9 nuclease has been adapted to target epigenomic modifications, but a detailed description of the parameters of such synthetic epigenome remodellers is still lacking. Here we describe a Cas9-based histone deacetylase (HDAC) and the design principles required to achieve locus-specific histone deacetylation. We assess its range of activity and specificity, and analyse target gene expression in two different cell types to investigate cellular context-dependent effects. Our findings demonstrate that the chromatin environment is an important element to consider when utilizing this synthetic HDAC.

[1] Department of Genetics, University of Pennsylvania School of Medicine, Philadelphia, Pennsylvania 19104, USA. Correspondence and requests for materials should be addressed to Z.Z. (email: zhaolan@mail.med.upenn.edu).

The epigenome is an essential element for the establishment and maintenance of cellular identity, acting in part through post-translational histone modifications that alter chromatin architecture and the transcriptional landscape of a cell[1]. High-throughput sequencing studies have uncovered a number of these histone modifications in a wide variety of biological settings, complementing a growing body of work describing the inheritance and transmission of such marks over time and across generations[2–4]. These studies have prompted a need for locus-specific epigenomic-editing methods to establish causality between the observed epigenetic features and biological phenotypes.

Adaptation of the bacterial genome-editing system, CRISPR-Cas9, has enabled targeted epigenomic editing through the coupling of a catalytic-inactive form of the RNA-guided Cas9 endonuclease to various chromatin-remodelling effectors[5–9]. Applicability of this approach in modifying histone marks has been demonstrated in studies describing fusions of dCas9 to the histone demethylase LSD1 (ref. 8) and histone acetyltransferase p300 (ref. 7), which decreased dimethylation at H3K4 and increased acetylation of H3K27 at target loci, respectively. A method for directed histone deacetylation, however, has yet to be developed. Furthermore, the effects of genomic features, such as locus-specific transcription and chromatin state, on the optimal positioning and efficacy of such dCas9 effectors have yet to be characterized.

To address this need, we designed a synthetic, programmable HDAC comprised of dCas9 and a histone deacetylase. Of those that have been identified in mammalian cells, the class I HDAC family consisting of HDACs 1, 2, 3 and 8 are ubiquitously expressed, primarily localized in the nucleus, and show high enzymatic activity towards histone substrates[10]. We chose HDAC3 to construct our synthetic HDAC, due to its well-established role in modulating transcription at promoters[11–13]. Although its effect on gene expression has largely been defined as repressive, it has also been shown to associate with active promoters enriched for RNA Pol II, H3K27ac and H3K4me3 (refs 11,14), as well as active transcription[15]. Given its role and localization, we hypothesized that targeting HDAC3 to endogenous promoters could induce histone deacetylation and alter gene expression. To test this, we fused full-length human HDAC3 to dCas9 (Fig. 1a) and measured its ability to modulate the transcription of several candidate genes. We sought to determine positions within the promoter required for dCas9-HDAC3 activity—varying both proximity to genomic sequences associated with histone acetylation and the transcription start site—and assessed dCas9-HDAC3's range of deacetylation activity. We demonstrate that the dCas9-HDAC3 fusion protein functions as a synthetic histone deacetylase and modulates gene expression when positioned appropriately in the promoter. We also show that elements such as target gene transcription level and acetylation status, gRNA positioning and gRNA dosage impact impacts dCas9-HDAC3 activity.

## Results

**Selection of target loci.** Chromatin features and endogenous transcription levels have been proposed to influence the efficacy of CRISPR-dCas9 at target genes[16,17]. We therefore chose to test our dCas9-HDAC3 system on several genes of varying expression levels and histone acetylation states at their respective promoter regions. To select our target loci, we first profiled genome-wide mRNA expression by RNA-seq, and surveyed genome-wide histone acetylation by performing chromatin immunoprecipitation followed by high-throughput DNA sequencing (ChIP-seq) for H3K27ac, a chromatin mark that is

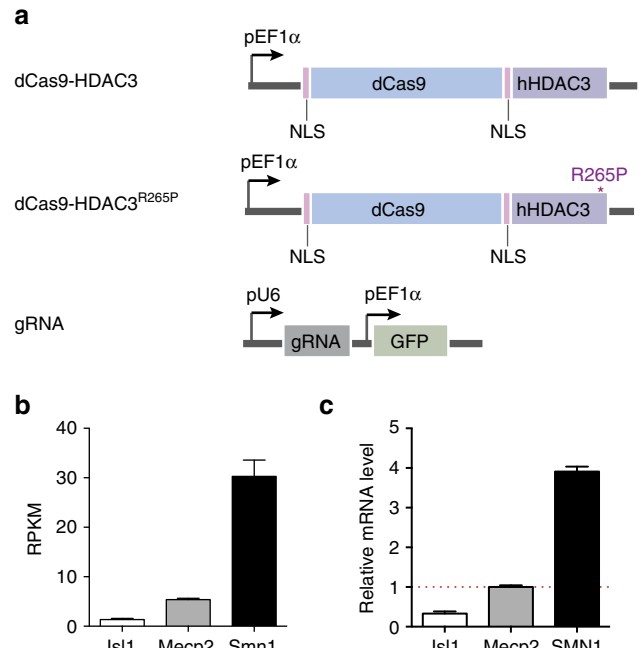

**Figure 1 | Design of Cas9-based HDAC system.** (**a**) Schematic of dCas9 fusion constructs, dCas9-HDAC3 and dCas9-HDAC3[R265P], and gRNA construct. (**b**) RPKM values for *Isl1*, *Mecp2* and *Smn1* in murine N2a cells as determined by RNA-seq (n = 2 biological replicates; error bars, s.e.m.). (**c**) *Isl1*, *Mecp2* and *Smn1* expression, relative to *Mecp2* (red dashed line) in N2a cells (n = 3 biological replicates; error bars, s.e.m.).

associated with active transcription[18] and HDAC3 (ref. 19), in the murine neuroblastoma cell line, Neuro-2a (N2a). We selected three genes to modulate, *Smn1*, *Mecp2* and *Isl1*, based on the dissimilarity of their expression levels, which we determined by RNA-seq (Fig. 1b) and validated by quantitative reverse-transcription PCR (RT–PCR) (Fig. 1c), and H3K27ac enrichment at their promoters (Fig. 2a,c,e). To determine the ideal positions for HDAC-dependent regulation of gene expression at promoter regions, we designed several gRNA target sites across the proximal promoter of each gene and transiently co-transfected each with our dCas9-HDAC3 fusion construct. Previous studies using dCas9-modified activators and repressors have reported an optimal window of gRNA positioning proximate to the transcriptional start site (TSS) and a decrease in efficacy as dCas9 is directed further away[16,20]. Additionally, dCas9:gRNA complexes bound to sequences downstream of the TSS have been shown to impair transcription, likely due to its hindrance of transcriptional machinery[21]. However, whether these rules apply broadly to other chimeric dCas9 effectors, including our dCas9-HDAC3 fusion protein, is unknown. We addressed this by testing gRNAs complementary to three different regions of each promoter of our target genes: (1) within 100 bp of the TSS; (2) 100–300 bp upstream of the TSS; (3) and 400–600 bp upstream of the TSS. To substantiate the specificity of our fusion enzyme, we created a construct containing a single inactivating mutation within HDAC3's catalytic domain (R265P (ref. 22)) and a non-targeting scrambled gRNA to co-transfect with both wild-type dCas9-HDAC3 and mutant dCas9-HDAC3[R265P] constructs (Fig. 1a) as controls in each experiment.

**Modulation of gene expression by transfected dCas9-HDAC3.** We first targeted dCas9-HDAC3 to three sites in the promoter of *Smn1* (gRNA-1: +11 bp; gRNA-2: −325 bp; gRNA-3: −463 bp),

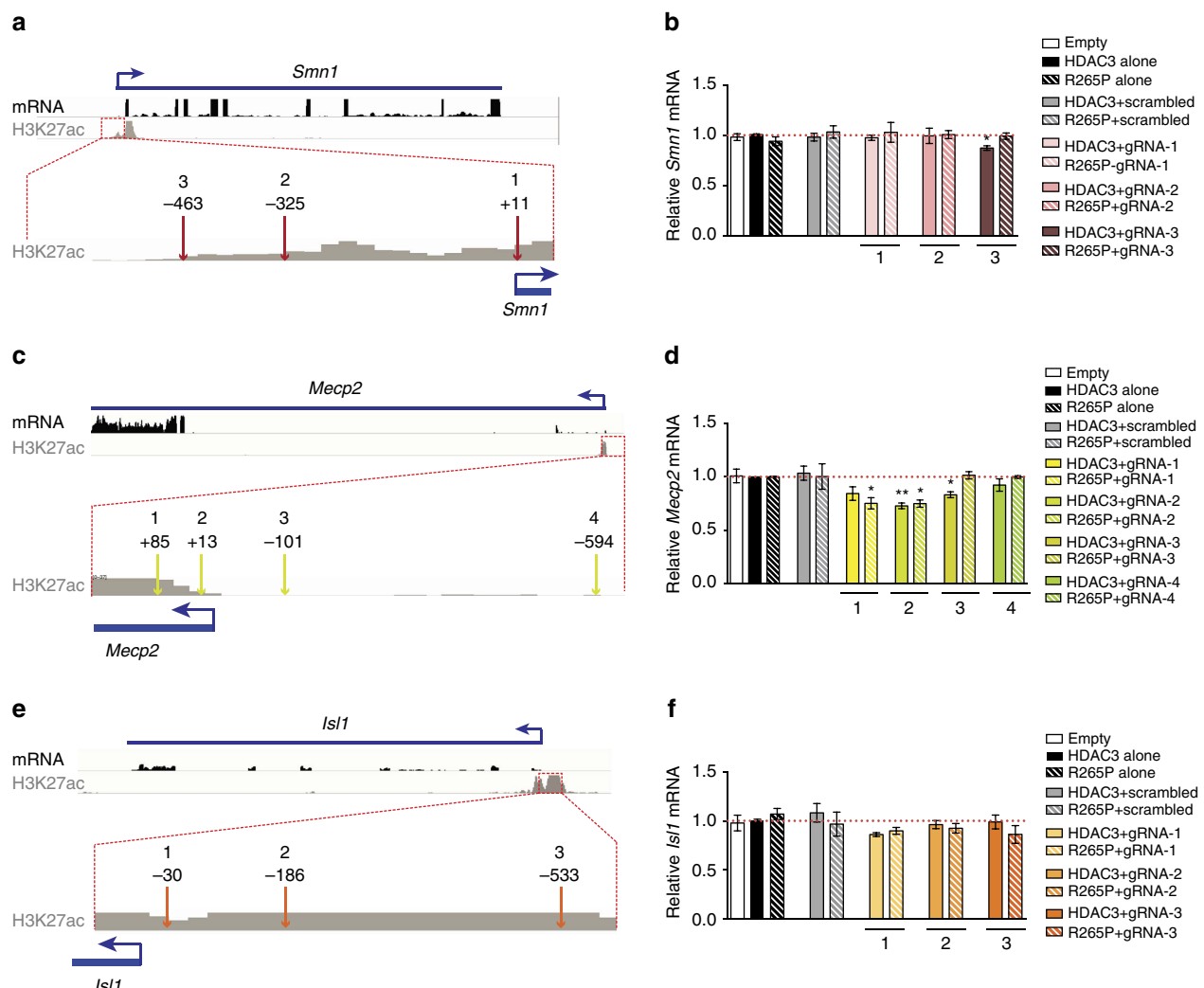

**Figure 2 | Positional effects of dCas9-HDAC3 on gene expression at three different loci.** (**a**) RNA-seq and H3K27ac ChIP-seq reads mapped to a region encompassing the murine *Smn1* gene. Magnified insets of H3K27ac enrichment at the *Smn1* promoter locus with approximate locations of each gRNA-targeted site are displayed below. (**b**) Quantitative RT–PCR measurements of relative mRNA levels of *Smn1* from N2a cells co-transfected with each gRNA and the indicated dCas9 proteins (relative to cells transfected with dCas9-HDAC3 alone; red dashed line) *$P < 0.05$; $n = 7$ biological replicates; error bars, s.e.m. (**c**) Schematic depiction of a region encompassing the murine *Mecp2* gene showing RNA-seq and H3K27ac tracks. Magnified insets of H3K27ac enrichment at the promoter locus with approximate locations of each gRNA-targeted site are displayed below. (**d**) RT–PCR measurements of mRNA levels of *Mecp2* from N2a cells co-transfected with each gRNA and the indicated dCas9 proteins (relative to cells transfected with dCas9-HDAC3 alone; red dashed line) *$P < 0.05$; **$P < 0.01$; $n = 7$ biological replicates; error bars, s.e.m. (**e**) RNA-seq and H3K27ac ChIP-seq reads mapped to a genomic region encompassing murine *Isl1*. Magnified insets of H3K27ac enrichment at the *Isl1* promoter locus with approximate locations of each gRNA-targeted site are displayed below. (**f**) RT–PCR measurements of mRNA levels of *Isl1* from N2a cells co-transfected with each gRNA and the indicated dCas9 proteins (relative to cells transfected with dCas9-HDAC3 alone; red dashed line) $n = 10$ biological replicates; error bars, s.e.m. All data were analysed using one-way ANOVA with Dunnett post-test.

which was determined by RNA-seq to be highly expressed in N2a cells (mean RPKM = 30.27) (Figs 1b and 2a). H3K27ac profiling by ChIP-seq showed two peaks at the *Smn1* promoter locus that extended to a region ∼500 bp upstream of the TSS. gRNA-3 targeted a region with low H3K27ac enrichment (9.36, normalized signal), near the 5′ end of H3K27ac, while regions complementary to gRNAs-1 and -2 located closer to the TSS exhibited higher levels of H3K27ac enrichment (31.12 and 14.16, respectively) (Fig. 2a). Co-transfection of each individual gRNA with either wild-type or mutant dCas9-HDAC3 revealed that only N2a cells expressing gRNA-3 and dCas9-HDAC3 show decreased *Smn1* expression relative to dCas9-HDAC3 alone ($P = 0.0176$; Dunnett's test) (Fig. 2b). Importantly, this alteration in *Smn1* was not observed when gRNA-3 was co-expressed with

dCas9-HDAC3$^{R265P}$, indicating that HDAC3's enzymatic activity is required for its repressive effect at this site (Fig. 2b). Moreover, both fusion proteins had no effect on *Smn1* transcription compared to non-transfected cells when expressed alone or with a non-targeting scrambled gRNA (Fig. 2b), suggesting that the effect we observed on *Smn1* expression represents a locus-specific effect of our CRISPR-based HDAC. These data demonstrate that dCas9-HDAC3 can regulate *Smn1* transcription, but that this regulatory function depends on its targeting location in the *Smn1* promoter and is ineffective at regions with high enrichment of H3K27ac at its binding location.

We next designed gRNAs targeting sites in a region of the *Mecp2* locus (mean RPKM = 5.365) (Fig. 1b) that has been shown to possess promoter activity in neurons and cultured cell lines[23,24].

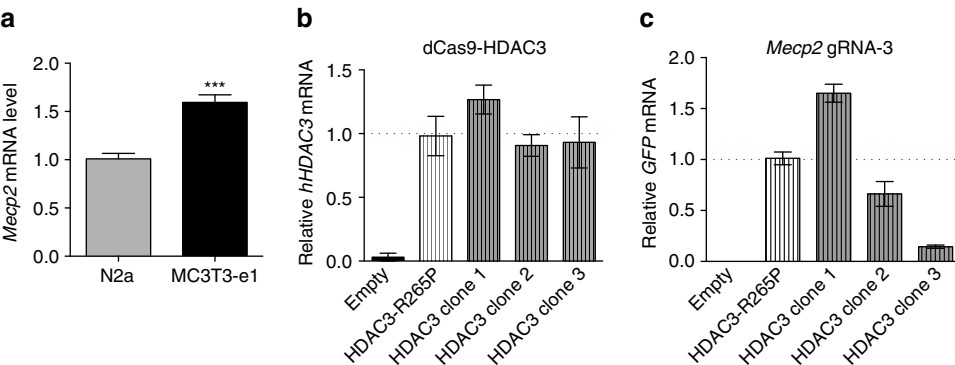

**Figure 3 | Generation of dCas9-HDAC3 and dCas9-HDAC3$^{R265P}$ MC3T3-e1 clonal cell lines targeting the *Mecp2* promoter.** (a) RT–PCR analysis of *Mecp2* expression in MC3T3-e1 cells relative to that of N2a cells. ***$P < 0.001$; $n = 5$ biological replicates; error bars, s.e.m. (**b**) Relative mRNA levels of human HDAC3 and (**c**) GFP in non-infected 'empty' murine MC3T3-e1 cells and MC3T3-e1-derived clonal cell lines stably expressing dCas9 fusion proteins and gRNA-3.

H3K27ac enrichment in this region was found to be positioned just downstream of the annotated TSS. Accordingly, we designed two gRNAs targeting H3K27ac enrichment downstream of the TSS (gRNA-1: +85; gRNA-2: +13) and two others complementary to regions upstream (gRNA-3: −101; gRNA-4: −594) (Fig. 2c). We then compared the regulatory function of dCas9-HDAC3 with each individual gRNA. Co-transfection of either dCas9-HDAC3 or dCas9-HDAC3$^{R265P}$ alone, or when co-expressed with a non-targeting scrambled gRNA, did not alter *Mecp2* expression compared to non-transfected cells (Fig. 2d). However, when directed to regions downstream of the TSS harbouring H3K27ac enrichment (gRNAs-1 and -2), we observed non-specific repression of *Mecp2* transcription (Fig. 2d). This decrease, independent of HDAC3's enzymatic activity, is likely due to steric inhibition of transcription and is consistent with previous reports of CRISPR interference (CRISPRi) by other Cas9-fusion proteins targeting regions downstream of the TSS[21,25].

Despite the fact that gRNA-3 (−101) targeted a region devoid of H3K27ac enrichment, we observed significant downregulation of *Mecp2* when it was co-expressed with dCas9-HDAC3 ($P = 0.0331$; Dunnett's test), an effect that was abolished in the HDAC3$^{R265P}$ mutant (Fig. 2d). In contrast, binding of either fusion protein had no significant effect when recruited further upstream of the *Mecp2* TSS (gRNA-4), suggesting this position exceeds the range of its activity (Fig. 2d). These results not only confirm dCas9-HDAC3's ability to repress gene transcription, but also indicate a limited range of efficacy that extends 101 bp beyond H3K27ac enrichment. Moreover, our findings support previous reports of Cas9-mediated impedance of transcription and indicate that dCas9-HDAC3 must be directed upstream of the TSS for catalytic-specific effects on transcription.

We next targeted the promoter of *Isl1*, which was found by RNA-seq to be lowly expressed in N2a cells (mean RPKM = 1.339) (Figs 1b and 2e). H3K27ac ChIP-seq showed high levels of enrichment spread across the promoter region and at each of our gRNA-binding sites (gRNA-1: −30; gRNA-2: −186; gRNA-3: −533, relative to TSS) (Fig. 2e). Expression of dCas9-HDAC3 or HDAC3$^{R265P}$ alone, or when co-transfected with a scrambled gRNA (Fig. 2f) did not impact *Isl1* expression. Notably, dCas9-HDAC3 targeting at each of the tested gRNA sites also failed to significantly alter transcription of *Isl1* compared to dCas9-HDAC3 alone (Fig. 2f). Thus, the effect of dCas9-HDAC3 appears to be context dependent; at promoter regions with high levels of H3K27ac, the repressive effect of the synthetic HDAC is limited. Taken together, our results from positioning dCas9-HDAC3 at similar locations in the promoters of three genes of varying

expression demonstrate a capacity for dCas9-HDAC3 to reduce gene expression when targeted appropriately in the promoter. Furthermore, we found that dCas9-HDAC3-mediated transcriptional regulation depends on local chromatin context, as its repressor activity was only observed at gRNA locations at or near the tail end of H3K27ac enrichment. These data indicate that dCas9-HDAC3 functions optimally when bound adjacent to H3K27ac marks in the promoter, thereby providing a positioning guideline when designing gRNA target sites in future studies.

The observed effects of dCas9-HDAC3 activity on gene transcription were relatively modest, indicating an inefficiency of transient transfections or weak effects of lone histone acetylation modulation on gene expression. The simultaneous expression of multiple gRNAs targeting a single promoter has been found to synergistically increase the effects of dCas9-mediated gene activation[26]. To examine whether gRNA multiplexing could augment dCas9-HDAC3 activity, we co-transfected N2a cells with either dCas9-HDAC3 or dCas9- HDAC3$^{R265P}$ and gRNAs that targeted the same promoter. To avoid the confounding effects of dCas9-mediated steric inhibition of transcription, only those gRNAs complementary to regions upstream of the TSS were co-expressed. For each locus tested (*Smn, Mecp2* and *Isl1*), co-transfection of two–three gRNAs in N2a cells did not result in any observable effects on transcription (Supplementary Fig. 1). In fact, any dCas9-HDAC3-mediated effect that we observed with a single gRNA (gRNA-3 for *Smn*; gRNA-3 for *Mecp2*) was abrogated when co-expressed with other gRNAs. This outcome could be a consequence of redirecting dCas9-HDAC3 from its optimal deacetylation position to other gRNA locations, or indicate impedance of its activity when dCas9-HDAC3 is also bound elsewhere in the promoter, thus reducing its effect on gene expression.

**Generating clonal cell lines expressing dCas9-HDAC3.** Having ascertained effective targeting sites for dCas9-HDAC3, we next examined additional parameters required for dCas9-HDAC3-mediated epigenomic editing. We chose to focus on one gene, *Mecp2*, to characterize the effect of dCas9-HDAC3. The expression of *Mecp2* must be precisely controlled, as increased and decreased levels of MeCP2 lead to the neurodevelopmental disorders MeCP2 Duplication Syndrome and Rett Syndrome[27], respectively. Experiments were performed in MC3T3-e1 pre-osteoblasts, which we found to express higher levels of *Mecp2* than N2a cells (Fig. 3a). This disparity in *Mecp2* expression between the two cell lines indicated distinct chromatin landscapes at this locus and allowed us

to investigate the efficacy of dCas9-HDAC3 in a different chromatin context.

Using gRNA-3, which repressed *Mecp2* transcription when co-expressed with dCas9-HDAC3, we first addressed the effects of gRNA dosage on dCas9-HDAC3 activity. Previous studies using dCas9-based epigenomic remodellers have primarily relied on transient transfections of their dCas9 and gRNA constructs, leading to variable levels of expression that have made dosage problematic to control in each cell. To overcome this impediment, we created multiple clonal cell lines stably expressing the dCas9-HDAC3 fusion proteins and *Mecp2* gRNA-3. MC3T3-e1 cells were infected with lentiviruses carrying either dCas9-HDAC3 or the inactive dCas9-HDAC3$^{R265P}$ followed by viral delivery of a construct containing both gRNA-3 and GFP (Fig. 1a). Cells were sorted for GFP fluorescence by fluorescence-activated cell sorting (FACS) and singly plated into a 96-well plate. Each individual cell gave rise to a clonal cell line that was propagated and tested for integration of human HDAC3 and GFP (Fig. 3b,c).

We first selected for clones expressing relatively uniform levels of the dCas9 fusion constructs (Fig. 3b) to prevent any confounding effects of varying dCas9-HDAC3 expression. Three dCas9-HDAC3 clonal lines, each containing varying amounts of gRNA-3 (HDAC3 clone 1: high; HDAC3 clone 2: intermediate; and HDAC3 clone 3: low, as determined by *GFP*), and one dCas9-HDAC3$^{R265P}$ clone (Fig. 3c), were selected to determine the effects of varying gRNA dosage for dCas9-HDAC3 function. As acetylation of H3K27 has been reported to associate with HDAC3 binding[19], we used enrichment of this histone mark as a measurement of dCas9-HDAC3-mediated deacetylation activity. To quantify targeted histone deacetylation in our clonal cell lines, chromatin immunoprecipitation (ChIP) for H3K27ac was performed in each clonal line followed by quantitative-PCR (ChIP-qPCR) using primers at regions immediately adjacent to gRNA-3. An amplicon encompassing a region 2 kb upstream of the TSS was used as a negative control to ensure specificity of our ChIP experiments (Fig. 4a).

**Modulation of H3K27ac at the *Mecp2* locus in clonal cells**. We found a significant loss of H3K27ac enrichment in the genomic regions immediately surrounding the gRNA-targeted site in both the high (HDAC3 clone 1) and intermediate (HDAC3 clone 2) gRNA-expressing clones relative to dCas9-HDAC3$^{R265P}$-transduced cells (Fig. 4b), in contrast to unaltered H3K27ac in the low gRNA-3-expressing clone (HDAC3 clone 3), suggesting a minimum level of gRNA expression is likely needed to achieve H3K27 deacetylation at this locus. Importantly, H3K27ac enrichment in 'empty' non-infected MC3T3-e1 cells was found to be comparable to that of the dCas9-HDAC3$^{R265P}$ clone (Fig. 4b), indicating that the enzymatic activity of HDAC3 is required for the deacetylation of H3K27ac at this locus.

Next, we performed ChIP-seq for H3K27ac in the clonal cell lines, R265P and HDAC3 clones 1 and 2, to assess the genome-wide effects of varying gRNA-3 expression on H3K27ac and to determine the range and specificity of dCas9-HDAC3 activity. We found that H3K27ac enrichment at the *Mecp2* promoter locus in dCas9-HDAC3$^{R265P}$ clonal cells occupied a larger genomic region than what was observed in N2a cells at this locus, extending past the gRNA-3 targeting site (Fig. 4c). We then subtracted the normalized H3K27ac uniquely mapped reads (RPM) in each clone from the normalized H3K27ac reads in dCas9-HDAC3$^{R265P}$-expressing cells across a 4 kb region centering around the *Mecp2* TSS. We observed a decrease in H3K27ac enrichment across the *Mecp2* promoter in both dCas9-HDAC3 clones relative to the dCas9-HDAC3$^{R265P}$ clone, and found that

this reduction spanned the entire H3K27ac peak (Fig. 4d). Notably, this effect was augmented in cells expressing higher amounts of gRNA-3 (HDAC3 clone 1) than in cells expressing a lower level of gRNA (HDAC3 clone 2) (Fig. 4d). These results not only confirm the ability of dCas9-HDAC3 to function as a targeted histone deacetylase, but also show that increased expression of gRNA results in a greater deacetylation of H3K27ac.

CRISPR-Cas9 proteins have been reported to tolerate mismatches in gRNA sequences and generate off-target effects[28,29]. To evaluate the specificity of dCas9-HDAC3, we next analysed the H3K27ac ChIP-seq profiles of dCas9-HDAC3 c1 and dCas9-HDAC3$^{R265P}$ at 17 different genomic loci harbouring predicted gRNA-3-binding sites containing up to four mismatches (Supplementary Table 3). This assessment allowed us to measure the relative amount of H3K27ac at potential dCas9-HDAC3 off-target-binding sites. Remarkably, we did not detect a significant decrease in H3K27ac at any of the tested sites (Fig. 4e). Together, these results suggest that the decrease in H3K27ac enrichment we observed in the dCas9-HDAC3 clones is due to targeting of the fusion protein at the *Mecp2* promoter and indicate minimal off-target deacetylation of H3K27.

Having observed a decrease in H3K27ac enrichment in two of the dCas9-HDAC3 clones relative to dCas9-HDAC3$^{R265P}$-expressing cells, we next examined *Mecp2* expression in all of our osteoblastic clonal lines. Remarkably, we found a significant upregulation of *Mecp2* in both of the higher gRNA-3-expressing clones relative to dCas9-HDAC3$^{R265P}$. In contrast, *Mecp2* expression was unaltered in dCas9-HDAC3 clone 3, which expressed the lowest level of gRNA, and in 'empty' non-infected cells (Fig. 4f). These results suggest that dCas9-HDAC3 promotes gene transcription at the *Mecp2* promoter in MC3T3-e1 pre-osteoblasts, contrary to its repressive function of *Mecp2* in transiently transfected N2a neuroblastoma cells. The opposing effects on *Mecp2* observed in these different cell lines, one neural in origin and the other bone derived, support previous studies that have demonstrated HDAC3's ability to bidirectionally regulate transcription[11,14], a function that may be influenced by local chromatin state.

## Discussion

We conclude that the dCas9-HDAC3 fusion protein can function as a synthetic histone deacetylase with just a single gRNA and modulate gene expression when positioned appropriately in the promoter. gRNA positioning was found to be critical for dCas9-HDAC3 activity and influenced by H3K27ac enrichment. Previous studies using dCas9 fused to transcriptional activators and repressors have described a window of optimal gRNA targeting dependent on TSS position. Our results suggest that such a TSS-proximal window is less relevant for dCas9-HDAC3. Instead, we found that the location of endogenous histone acetylation is a critical element for dCas9-HDAC3 function. We show that only gRNAs located adjacent to H3K27ac marks promoted a dCas9-HDAC3-dependent effect on transcription. Moreover, dCas9-HDAC3 at this position catalysed the deacetylation of H3K27 across an ∼1 kb region at the *Mecp2* locus across the entire H3K27ac peak, indicating that dCas9-HDAC3's range of activity extends beyond its immediate-binding site. These results indicate that chromatin features, such as histone acetylation, must be considered when designing gRNAs for dCas9-HDAC3 activity and may relate to why certain loci have been found to be less responsive to modulation by other dCas9 effectors. Furthermore, we show that gRNAs designed at sites downstream of the TSS can impede transcription independently of HDAC3's catalytic function. Of note, this phenomenon was

not observed at *Smn1*, a highly expressed gene. This is likely due to high occupancy of transcription machinery at the TSS that may compete with or impair dCas9-HDAC3 binding or may involve other chromatin features that were not investigated in this study. gRNAs located further from H3K27ac enrichment at the promoter, however, did not impart a discernible effect on target gene expression, indicating limitations in the range of dCas9-HDAC3 catalytic activity.

Our study also measured the effect of gRNA dosage on dCas9-HDAC3 function, taking advantage of transduced cell lines in which every cell in each line expresses homogenous levels of dCas9-HDAC3 and gRNA. Our observations of H3K27 deacetylation in both the high- and intermediate-gRNA-expressing clones, which positively correlated with their expression of gRNA-3, demonstrates a dose-dependent effect of gRNA on this system. These results suggest that titrating gRNA amounts can be used as a strategy to modulate dCas9-HDAC3 activity.

Our findings also support previous reports of HDAC3's ability to promote or repress gene transcription[11,14,15,19]. We demonstrate that the chromatin context is a critical factor specifying HDAC3's role and show opposing effects of HDAC3-mediated gene regulation in two different cell types. The mechanisms by which HDAC3 functions to reduce gene expression in one cell type and augment it in another, however, have yet to be determined. We postulate that HDAC3 may recruit different effectors to either upregulate or downregulate transcription at the promoter depending on the local chromatin environment. Indeed, comparison of H3K27ac profiles of N2a cells and our dCas9-HDAC3 clones revealed a size discrepancy in H3K27ac occupancy at the *Mecp2* promoter with a correlative increase in *Mecp2* expression in MC3T3-e1 cells relative to N2a cells (Fig. 3a). Despite the observed target-site-specific reduction in H3K27ac at the *Mecp2* promoter by dCas9-HDAC3 recruitment, we speculate that the observed increase in *Mecp2* expression in the MC3T3-e1 clonal cells may also occur by other means, including the deacetylation of additional histone marks and non-histone substrates that are endogenously bound at this locus[15]. Examination of additional histone marks and epigenetic factors may elucidate dCas9-HDAC3's opposing roles in gene transcription.

Other approaches to manipulate epigenetic states in a locus-specific manner utilize transcription activator-like effectors (TALEs) and zinc-finger proteins (ZFPs), which are designed to bind specific loci of interest[30]. An adapted TALE system was used to recruit the transcriptional regulators NCoR and Sin3a[31], which form complexes with HDAC3 (refs 13,19) and HDAC1/2 (ref. 32), respectively, thereby indirectly site-recruiting HDACs to target epigenetic modifications. The same system also demonstrated an ability to repress transcription and reduce H4K8ac when fused to HDAC8 (ref. 31). Interestingly, these

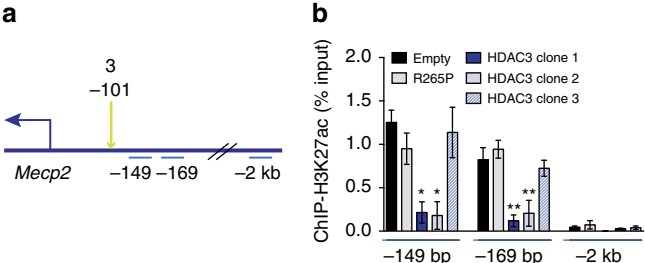

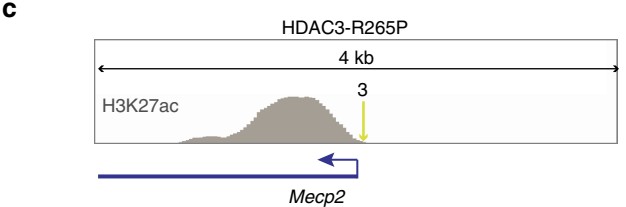

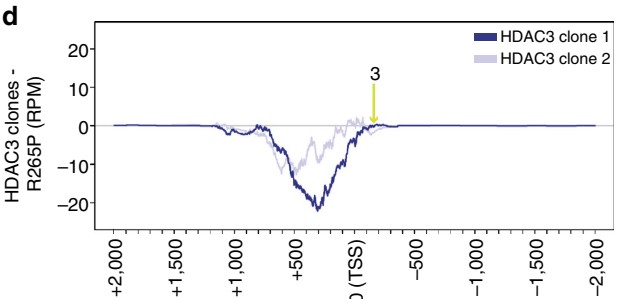

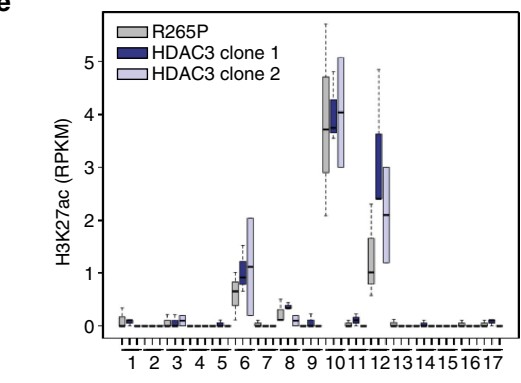

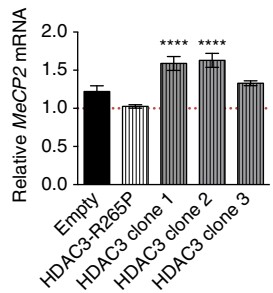

**Figure 4 | Assessing gRNA dosage and dCas9-HDAC3 range of activity at the *Mecp2* locus.** (**a**) The *Mecp2* promoter locus is schematically depicted with the approximate locations of gRNA-3 and the ChIP-qPCR amplicon regions (in blue). (**b**) H3K27ac ChIP-qPCR enrichment (relative to dCas9-HDAC3$^{R265P}$) at regions in the *Mecp2* promoter depicted in Fig. 2c. One-way ANOVA with Dunnett post-test; * $P<0.05$; **$P<0.01$; $n=4$ biological replicates; error bars, s.e.m. (**c**) H3K27ac enrichment at a genomic locus encompassing a 4 kb region around the *Mecp2* TSS in dCas9-HDAC3$^{R265P}$ clonal cells. Green arrow indicates the position of gRNA-3. (**d**) Genomic region centred on the *Mecp2* TSS (0 = TSS) depicting the subtraction of the normalized H3K27ac uniquely mapped reads (RPM) in each dCas9-HDAC3 clone (clone 1: $n=3$ biological replicates; clone 2: $n=2$ biological replicates) from the normalized H3K27ac reads (RPM = reads per million) in the dCas9-HDAC3$^{R265P}$ clone ($n=3$ biological replicates). (**e**) H3K27ac RPKM values for the dCas9-HDAC3$^{R265P}$ clone (R265P), dCas9-HDAC3 clone 1 (HDAC3 clone 1) and dCas9-HDAC3 clone 2 (HDAC3 clone 2) at 17 potential off-target sites for *Mecp2* gRNA-3 in the genome. (**f**) Relative mRNA expression of *Mecp2* determined by RT–PCR by the indicated dCas9-HDAC3 clones co-expressing *Mecp2* gRNA-3 (relative to dCas9-HDAC3$^{R265P}$ also co-expressing *Mecp2* gRNA-3) One-way ANOVA with Dunnett post-test; ****$P<0.0001$; $n=7$ biological replicates; error bars, s.e.m.

TALE-modified effectors showed varying effects on gene repression across different genes in two different cell types, suggesting a similar functional dependency on gRNA positioning and/or local chromatin state that was observed with dCas9-HDAC3 in this study. Similarly, ZFPs fused to the catalytic domain of histone methyltransferase G9a have been reported to increase enrichment of H3K9me2 and repress transcription in a site-specific manner[33,34]. Although the readily programmable nature of CRISPR-Cas9 facilitates its usability, we surmise that combining dCas9 effectors with other engineered ZFPs and TALEs may be necessary to reproduce various chromatin features at loci of interest.

The dCas9-HDAC3 system is a unique addition to the CRISPR-dCas9 epigenome-editing toolbox, and provides a method in which histone deacetylation of genomic loci associated with various developmental and disease states can be directly interrogated. Future work is needed to determine if the dCas9-HDAC3 system functions broadly at other gene promoters and can also alter gene expression when positioned at other regulatory genomic elements, such as enhancers or insulators. Its ability to deacetylate additional histone marks and bidirectionally regulate gene expression also warrant investigation. Moreover, an expanded examination of chromatin state that include additional epigenetic features may elucidate why certain gRNAs work more optimally with dCas9-modified enzymes than others, and are necessary for future application of this synthetic HDAC in other cell types and systems. Together with other already established CRISPR-based epigenome remodelling tools, dCas9-HDAC3 represents a powerful technology that can be used to further our knowledge of gene regulation and the functional role of epigenomic modifications.

## Methods

**Plasmid constructs.** The lentiviral vector containing the dCas9-HDAC3 fusion protein was adapted from the already described pLenti-dCas9$^{VP64}$ plasmid (Addgene, Plasmid #61425). The VP64 effector domain was removed from the plasmid by digestion with BamHI and BsrGI and full-length human HDAC3 (PCR amplified from Addgene plasmid #13819 with BamHI/BsrGI restriction sites) was cloned into the pLenti-dCas9$^{VP64}$ backbone. The inactive R265P point mutant was created by nucleotide substitution in the original Addgene HDAC3 vector using QuikChange Site-Directed Mutagenesis Kit (Agilent Technologies, cat. #200519) and subcloned into pBluescript (kind gift from Dr Doug Epstein). The resulting full-length HDAC3$^{R265P}$ was amplified with BamHI and BsrGI restriction enzyme ends and cloned into BamHI/BsrGI digested pLenti-dCas9$^{VP64}$.

gRNA expression vectors were generated by annealed oligo cloning using BsmBI of Addgene plasmid #57822. gRNA target sites were obtained by using the following gRNA design tools: CRISPR-Design (http://crispr.mit.edu/) and CRISPR-ERA (http://crispr-era.stanford.edu/). Only gRNAs with at least three mismatches to other sites in the genome other than the target site and with similar scores were used. Oligonucleotides for gRNA construction were obtained from Integrated DNA Technologies and their sequences are shown in Supplementary Table 1.

All PCR amplifications were carried out using Phusion Hi-Fidelity DNA Taq polymerase (NEB cat. #M0530S).

**Cell lines and transfection.** N2a and HEK293T cells were procured from the American Tissue Collection Center (ATCC, Manassas, VA, USA) and cultured in Dulbecco's modified Eagle's medium supplemented with 10% FBS and 1% penicillin/streptomycin. MC3T3-e1 cells (kind gift from Dr Eileen Shore at the Perelman School of Medicine) were cultured in alpha minimum essential medium supplemented with 10% FBS. Both cell lines were maintained at 37 °C and 5% CO$_2$. Transfections were performed in six-well plates using 1 μg of each respective dCas9 expression vector, 0.5 μg of individual gRNA expression vectors and Lipofectamine 2000 (Life Technologies, cat. #11668019) as per the manufacturer's instruction. For ChIP-qPCR experiments, MC3T3-e1 cells were plated in 15-cm dishes.

**Generation of stable MC3T3-e1 clonal cell lines.** Lentivirus production: HEK293T cells plated in 10-cm dishes seeded at ∼80% confluency were transfected with 12 μg lentiviral dCas9 fusion or gRNA constructs, 6 μg of the helper plasmids VSVG and Δ8.9 each, and Lipofectamine 2000 according to the manufacturer's instruction. Viral supernatant was collected 48 h post transfection, filtered with a 0.45 μM polyvinylidene difluoride filter (Millipore), aliquoted and stored at −80 °C.

Lentiviral transduction: MC3T3-e1 cells were transduced first with dCas9 fusion lentivirus via spinfection in six-well plates containing 2 ml of lentiviral supernatant media supplemented with 10 μg ml$^{-1}$ polybrene (Santa Cruz, cat. #sc-134220). Plates were centrifuged for 30 min at 1,000g, 37 °C. Media was immediately replaced with fresh alpha minimum essential media. Cells were propagated for 1 week before another round of infection with gRNA-3 lentivirus in the same manner as described above. Lentivirus-infected cells were then singly sorted according to GFP by FACS into 96-well plates using the BD FACS Aria II and expanded to generate clonal cell lines.

**Quantitative RT–PCR.** RNA was extracted with Trizol reagent (Invitrogen) and purified using the RNeasy MinElute Clean-up kit (Qiagen cat. #74204). Thousand nanogram of RNA was converted into cDNA using the High-Capacity cDNA Reverse Transcription Kit (Applied Biosystems, 4368814) and real-time PCR was performed using Taqman Gene Expression Assay probes purchased from Applied Biosystems and TaqMan Universal PCR Master Mix (Applied Biosystems, cat. #4304437). The following Taqman assay primer/probe sets were used for this study: Gapdh (Mm99999915_g1); Hprt (Mm03024075_m1); Mecp2 (Mm01193537_g1); HDAC3 (Hs00187320_m1); Smn1 (Mm00488315_m1); Isl1 (Mm00517585_m1) and a Taqman primer/probe set to detect GFP was designed as follows:

Forward: 5′-TGCTTGTCGGCCATGATATAG-3′ (Sense)
Reverse: 5′-GAACCGCATCGAGCTGAA-3′ (AntiSense)
Probe: 5′-ATCGACTTCAAGGAGGACGGCAAC-3′ (AntiSense)

Results were quantified on an ABI 7900 system. All RNA expression levels were normalized to Gapdh (for MC3T3-e1 experiments) or Hprt (for N2a experiments) using the ΔΔCT method.

**ChIP-qPCR and ChIP-seq.** Cells plated on 15-cm dishes were crosslinked in buffer containing 100 mM HEPES (pH 7.5), 100 mM NaCl, and 1 mM EDTA and 1 mM EGTA and for 5 min at room temperature, followed by quenching with 0.125 M glycine. Cells were lysed in cell lysis buffer (50 mM HEPES (pH 7.5) 140 mM NaCl, 1 mM EDTA, 1 mM EGTA, 0.25% Triton X-100, 0.5% NP-40, 10% glycerol) for 10 min on ice, and nuclei were collected and lysed in 10 mM Tris pH 8.0, 0.1% SDS, 1 mM EDTA and 1 mM EGTA. DNA shearing was performed on a Covaris S220 (12 min sonication, 5% duty factor). Buffer conditions were adjusted to contain 150 mM NaCl, 0.1% SDS and 0.5% Triton X-100 and chromatin was precleared with Protein A Dynabeads (Invitrogen). Immunoprecipitation was performed using 32 μl Protein A Dynabeads and antibodies against H3K27ac (Abcam ab4729), or preimmune rabbit IgG. Beads were washed twice with low-salt buffer (150 mM NaCl, 50 mM HEPES pH 7.5, 0.1% DOC, 1% Triton X-100, and 1 mM EDTA), once with high-salt buffer (500 mM NaCl, 0.1% DOC, 50 mM HEPES pH 7.5, 1% Triton X-100, and 1 mM EDTA), once with LiCl buffer (10 mM Tris-HCl pH 8.1, 250 mM LiCl, 0.5% NP-40, 0.5% DOC, 1 mM EDTA, and twice with TE. Chromatin was eluted with elution buffer (50 mM Tris-HCl (pH 8.0), 10 mM EDTA and 1% SDS, wt/vol) and reverse crosslinked overnight at 65 °C, followed by treatment with RNase A for 30 min at 42 °C and proteinase K for 3 h at 55 °C. DNA was extracted twice with phenol/chloroform and once with chloroform and ethanol precipitated. Quantitative ChIP was performed in duplicate on an ABI 7900 instrument using SYBR Green detection. Serial dilutions of input DNA were used to generate a standard curve for each primer pair. All ChIP-qPCR primers are listed in Supplementary Table 2.

ChIP-seq libraries were constructed using NEBNext ChIP-Seq Library Prep Kit reagents, Illumina ChIP Sample Prep adaptors, and deep sequenced on the Illumina HiSeq 2500 on Rapid Run Mode. FASTQ files for each ChIP-seq library was mapped to the mouse mm10 genome by Bowtie[ref PMID: 19261174] using the parameters of ' − v 2 –m 1', and only uniquely mapped reads were included in the downstream analysis.

**Statistical analyses.** Data values are represented as mean ± s.e.m. and were analysed in Excel (Microsoft) and Prism6 (GraphPad). Measurements were analysed for statistical significance using one-way analysis of variance (ANOVA) multiple comparison tests with Dunnett's post hoc tests unless otherwise stated. Statistical significance was set to $P < 0.05$.

**RNA sequencing and data analysis.** Samples collected for RNA sequencing were prepped with TruSeq Stranded mRNA Sample Prep Kit (Illumina) and deep sequenced on the Illumina HiSeq platform. FASTQ files for each RNA sequencing library was mapped to the mouse mm10 genome by STAR[ref PMID: 23104886] using the parameters of '—outFilterMultimapNmax 1 –outFilterMismatchNmax 3'. The read count of each gene was performed by in-house Perl programmes and was normalized by edgeR[ref PMID: 199103].

**Data availability.** Sequencing data can be found in NCBI GEO with the accession code GSE91043. Other data and reagents relevant to this study are available from the corresponding author on request.

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

## Acknowledgements

We thank the Flow Cytometry & Cell Sorting Facility at the University of Pennsylvania Perelman School of Medicine for their FACS services. We also thank the Next Generation Sequencing Core at the Perelman School of Medicine for their high-throughput sequencing services. This work is partially supported by the Brain Research Foundation, NIH R01MH091850, and a start-up fund from the University of Pennsylvania (Z.Z.). D.Y.K. is supported by the T32 Training Programme in Neurodevelopmental Disabilities (T32NS007413). Z.Z. is a Pew Scholar in biomedical science.

## Author contributions

D.Y.K. and Z.Z. were responsible for the conception, design and interpretation of experiments and wrote the manuscript. D.Y.K. and J.M.L. conducted experiments, and Y.-T.Z. analysed all next-generation sequencing data.

## Additional information

**Competing interests:** The authors declare no competing financial interests.

