## [Peer Review File · Nature Communications]

Reviewers' Comments:

Reviewer #1 (Remarks to the Author)

This interesting study describes a CRISPR-Cas-based approach for targeted Histone lysine deacetylation in cells.

Full length HDAC3 is fused to dCas and several genes are targeted for analysis of mRNA transcription regulation in N2A cells. Genes are chosen on basis of own RNA-Seq and Chip to cover range of expression and Ac levels. Positioning of gRNAs is varied to study positional influence. Then stable MC3T3-e1 cells are generated and gRNA dosage tested on basis of H3K27Ac Chip. Positional resolution of Ac is evaluated and off-target effects at 17 sites. Optimal system is again evaluated in stables for effect on mRNA levels, showing stronger effect than before in transiently transf. N2A (and adverse effects in both lines).

Overall the study contributes an interesting addition to the targeted epigenome editors toolbox and is thoroughly performed with the important controls.

The greatest weakness of the study is the weakness of the observed effects on mRNA expression if the constructs were to be used as mRNA modulators, in case of N2a barely significant.

Nevertheless, effects in H3K27Ac are stronger, showing that tools themselves work and thus data reflect low impact of Ac in mRNA levels. Nevertheless, it would be highly interesting - in particular given the ease of multiplexing with CRISPR - to see, if synergetic effects on mRNA modulation would be possible by using several gRNAs simultaneously and one example should be included in the manuscript. Moreover, I suggest to include all previous examples of targeted Sin3A/KRAB editors (be it ZFP or TALEs), since these provide an indirect way to recruit HDAC to chosen loci and thus are relevant state of the art for this paper. Most importantly, HDAC8 has previously been used in combination with TALEs for optical Ac control (Konerman et. al. Nature, 2013, showing stronger effects on mRNA levels in the Grm2 gene). Paper must be included in reference list and data have to be discussed in context with present data.

Reviewer #2 (Remarks to the Author)

The manuscript describes basic rules of applying dCas9-HDAC3 for directing targeted changes to the histone modifications and consequent gene regulation. Many of the analyses are clear and to the point, especially the last two figures (3&4). Statistical analyses seem reasonable. Design principles presented in the manuscript will be very useful for targeted epigenetic modifications using HDAC3. In addition, the context dependent effect of HDAC3 was very interesting observation, especially in figure 4. The study does not go far enough to resolve the predictability of the HDAC3 action; this may be a missed opportunity. Nonetheless, the authors presents nice set of data that will be valuable in designing targeted epigenetic modifiers.

One minor criticism of the paper is the relatively weak effects presented in figure 2. The magnitude of the effect is very small, when compared to the previously published work (or even to the results presented in figure 4). This is likely due to the transient plasmid transfection approach used in these studies, even though the vectors are suitable for viral packaging and transduction for a more potent delivery (as seen in other reports).

Response to referees:

Reviewer #1:

This interesting study describes a CRISPR-Cas-based approach for targeted Histone lysine deacetylation in cells. Full length HDAC3 is fused to dCas and several genes are targeted for analysis of mRNA transcription regulation in N2A cells. Genes are chosen on basis of own RNA-Seq and Chip to cover range of expression and Ac levels. Positioning of gRNAs is varied to study positional influence. Then stable MC3T3-e1 cells are generated and gRNA dosage tested on basis of H3K27Ac Chip. Positional resolution of Ac is evaluated and off-target effects at 17 sites. Optimal system is again evaluated in stables for effect on mRNA levels, showing stronger effect than before in transiently transf. N2A (and adverse effects in both lines).

Overall the study contributes an interesting addition to the targeted epigenome editors toolbox and is thoroughly performed with the important controls. The greatest weakness of the study is the weakness of the observed effects on mRNA expression if the constructs were to be used as mRNA modulators, in case of N2a barely significant. Nevertheless, effects in H3K27Ac are stronger, showing that tools themselves work and thus data reflect low impact of Ac in mRNA levels. Nevertheless, it would be highly interesting - in particular given the ease of multiplexing with CRISPR - to see, if synergetic effects on mRNA modulation would be possible by using several gRNAs simultaneously and one example should be included in the manuscript. Moreover, I suggest to include all previous examples of targeted Sin3A/KRAB editors (be it ZFP or TALEs), since these provide an indirect way to recruit HDAC to chosen loci and thus are relevant state of the art for this paper. Most importantly, HDAC8 has previously been used in combination with TALEs for optical Ac control (Konerman et. al. Nature, 2013, showing stronger effects on mRNA levels in the Grm2 gene). Paper must be included in reference list and data have to be discussed in context with present data.

We thank this reviewer for his/her appreciation of our study and for recognizing the value of our findings. We also appreciate their suggestion of multiplexing gRNAs to examine whether their simultaneous expression might synergistically improve dCas9-HDAC3's effect on gene transcription as has been observed with CRISPR-based activators. To address this, we co-expressed 2-3 gRNAs used in this study with either dCas9-HDAC3 or dCas9-HDAC3-R265P for each respective locus. To avoid the confounding effect of steric inhibition of transcription (as was observed when dCas9-HDAC3 or -R265P was positioned downstream of the TSS) we only chose to multiplex those gRNAs complementary to regions upstream of the TSS (Fig. 1).

Fig. 1. Multiplexing gRNAs with dCas9-HDAC3 does not result in synergistic repression of gene expression at tested loci. 48 hr co-transfection of a) *Smn* gRNAs 2 and 3; b) *Mecp2* gRNAs 3 and 4; c) and *Isl1* gRNAs 1, 2, and 3, with either dCas9-HDAC3 or dCas9-HDAC3-R265P do not significantly alter the expression of each respective gene relative to dCas9-HDAC3 alone. *Smn* (n= 6); *Mecp2* (n=6); *Isl1* (n=6).

For each locus tested (*Smn*, *Mecp2*, and *Isl1*), co-transfection of 2 to 3 gRNAs in N2a cells did not result in any observable effects on transcription (Fig. 1). In fact, any dCas9-HDAC3-mediated effect that we observed with single expression of gRNA (*Smn* gRNA-3 on *Smn*; *Mecp2* gRNA-3 on *Mecp2*) was abrogated when co-expressed with another gRNA. This may be due to the diverting of dCas9-HDAC3

from its optimal deacetylation position to other gRNA locations, or impedance of its activity when dCas9-HDAC3 is also bound elsewhere in the promoter, thus reducing its effect on gene expression. However, as most our gRNAs were designed to bind to positions approximately one nucleosome or more apart, we cannot rule out the possibility that synergistic effects may be seen with the co-expression of multiple gRNAs tiled closely around a promoter locus with dCas9-HDAC3. This figure has been added to our manuscript (Supplementary Fig. 1) and described on page 6, lines 20 to page 7, line 2 (highlighted).

We also thank the reviewer for advising us to include and discuss examples of epigenomic editing utilizing other epigenome remodelers (TALE-adapted-Sin3, NCoR, HDACs, etc.). We have now included those references in our manuscript on page 9, line 9-24 (highlighted).

Reviewer #2:

The manuscript describes basic rules of applying dCas9-HDAC3 for directing targeted changes to the histone modifications and consequent gene regulation. Many of the analyses are clear and to the point, especially the last two figures (3&4). Statistical analyses seem reasonable. Design principles presented in the manuscript will be very useful for targeted epigenetic modifications using HDAC3. In addition, the context dependent effect of HDAC3 was very interesting observation, especially in figure 4. The study does not go far enough to resolve the predictability of the HDAC3 action; this may be a missed opportunity. Nonetheless, the authors present nice set of data that will be valuable in designing targeted epigenetic modifiers.

One minor criticism of the paper is the relatively weak effects presented in figure 2. The magnitude of the effect is very small, when compared to the previously published work (or even to the results presented in figure 4). This is likely due to the transient plasmid transfection approach used in these studies, even though the vectors are suitable for viral packaging and transduction for a more potent delivery (as seen in other reports).

We thank this reviewer for recognizing the utility and significance of our findings. We agree entirely that the reported dual activator/repressor activity of HDAC3, which was recapitulated in our study even when fused to dCas9, provides an interesting context-dependent insight into HDAC3's role in gene regulation. We did not further delve into that finding in this study, which was meant to report general rules and guidelines of dCas9-HDAC3-mediated epigenomic editing; however we expect future studies will work to resolve the mechanisms of this observed duality on gene transcription, which may provide additional insight into HDAC3 function.

We also agree with the reviewer that the inefficiency of transient transfection of N2a cells may have mitigated dCas9-HDAC3's effect on gene expression, leading only to modest reduction of gene expression. This point is why we undertook the effort to create stable clonal cell lines via lentiviral infection, which resulted in stronger effects on *Mecp2* expression. Additionally, gene expression is regulated by numerous chromatin modifications apart from histone acetylation. It is conceivable that histone deacetylation by HDAC3 alone may not be sufficient to produce strong effects on gene transcription, at least those observable by 48-hour transient transfections. We envision the dCas9-HDAC3 system to be used beyond solely as a means of manipulating gene expression, to providing a much-needed method of interrogating the function of locus-specific histone acetylation marks. Such tools are necessary to establish causal relationships between histone modifications and observed cellular and biological phenotypes. We have now included this point in our manuscript (highlighted) on page 6, lines 20-22.

Reviewers' Comments:

Reviewer #1 (Remarks to the Author)

my comments have been fully addressed and i support publication in this revised form.

Reviewer #2 (Remarks to the Author)

The revised manuscript addresses all the prior concerns and criticisms of the reviewers and is suitable for publication.